# POISONING AND BACKDOORING CONTRASTIVE LEARNING

**Nicholas Carlini**
Google

**Andreas Terzis**
Google

## ABSTRACT

Multimodal contrastive learning methods like CLIP train on noisy and uncurated training datasets. This is cheaper than labeling datasets manually, and even improves out-of-distribution robustness. We show that this practice makes *backdoor* and *poisoning* attacks a significant threat. By poisoning just $0.01\%$ of a dataset (e.g., just 300 images of the 3 million-example Conceptual Captions dataset), we can cause the model to misclassify test images by overlaying a small patch. Targeted poisoning attacks, whereby the model misclassifies a particular test input with an adversarially-desired label, are even easier requiring control of $0.0001\%$ of the dataset (e.g., just three out of the 3 million images). Our attacks call into question whether training on noisy and uncurated Internet scrapes is desirable.

## 1 INTRODUCTION

*Contrastive learning* (Chopra et al., 2005; Hadsell et al., 2006) trains a model that projects a data distribution onto a lower-dimensional embedding space such that similar objects in the origin space are closer together in the embedding space than dissimilar objects (Chechik et al., 2010; Sohn, 2016; Oord et al., 2018; Wu et al., 2018). Significant advances over the last years have enabled self-supervised classifiers to achieve state of the art accuracy by training on noisy and uncurated datasets (Radford et al., 2021; Tian et al., 2021), which brings two significant benefits.

First, training on uncurated data is cheaper (Joulin et al., 2016). Compared to an estimated several million USD it cost to label the ImageNet dataset (Deng et al., 2009), contrastively trained models can train without expensive labeling efforts (Chen et al., 2020a). Further, because each image in ImageNet is required to contain one of just 1,000 different objects, there are large categories of images that can never be part of this supervised dataset (Jia et al., 2021). On the other hand, a contrastive model can learn on arbitrary images whether or not they have a suitable corresponding label in some dataset.

Second, training on noisy data improves robustness (Radford et al., 2021). Classifiers trained exclusively on ImageNet overfit the particular details of this training set (Recht et al., 2019; Hendrycks & Dietterich, 2019), and do not generalize to other test sets (Taori et al., 2020). Contrastive models trained on data scraped from the Internet exhibit impressive robustness properties; The contrastively trained CLIP (Radford et al., 2021) model is the first technique to show significant *effective robustness* on ImageNet-V2 (Recht et al., 2019; Taori et al., 2020).

**Contributions.** We make the case that training on unfiltered may be **un**desirable if even a tiny fraction of the data could be maliciously poisoned by an adversary. And this is likely the case: the data is scraped from the Internet (Jia et al., 2021) without *any* human review before it is passed to the learning algorithm (Radford et al., 2021; Jia et al., 2021; Tian et al., 2021). Thus, because these datasets are explicitly "noisy" (Jia et al., 2021) and "uncurated" (Tian et al., 2019), we argue the likelihood of at least one adversary is high.

We show that this adversary can mount powerful **targeted poisoning** (Biggio et al., 2012) and **backdoor** attacks (Gu et al., 2017; Chen et al., 2017) against multimodal contrastive models. A poisoning adversary introduces malicious examples into the training dataset so that the model will misclassify a particular input at test time as an adversarially-desired label. We then consider patch-based backdoors, where the adversary poisons a dataset so that the learned model will classify *any* input that contains a particular trigger-pattern as a desired target label.

We require no new technical ideas to poison or backdoor contrastively-trained models (Biggio et al., 2012; Gu et al., 2017; Chen et al., 2017)—although we must adapt existing techniques to this new

domain. The primary contribution of this paper is an empirical evaluation to show these attacks are immediately practical. Compared to prior backdooring attacks which require poisoning on average $1\%$ of training data for successful clean label attacks (Shafahi et al., 2018; Saha et al., 2021), we find that attacking multimodal contrastive models requires orders of magnitude fewer injections: just $0.01\%$ suffices for many of our backdoor attacks, or $0.0001\%$ for poisoning attacks.

## 2 BACKGROUND, NOTATION, AND RELATED WORK

### 2.1 POISONING AND BACKDOOR ATTACKS

In a poisoning attack (Biggio et al., 2012), an adversary modifies a benign training dataset $\mathcal{X}$ by injecting poisoned examples $\mathcal{P}$ to form a poisoned dataset $\mathcal{X}' = \mathcal{X} \cup \mathcal{P}$. When the victim runs the training algorithm $\mathcal{T}$ on the modified training dataset $X'$, they obtain a poisoned model $f_\theta \leftarrow \mathcal{T}(\mathcal{X}')$. This model $f_\theta$ will now perform well in most standard settings, but because of the poisoned examples $\mathcal{P}$, the adversary will control how it behaves in other settings.

We first consider *targeted poisoning* (Barreno et al., 2006; Biggio et al., 2012) where an adversary injects poisoned examples so that some input $x'$ will be misclasified as a desired target $y'$. Poisoning attacks exist for many tasks, including supervised (Biggio et al., 2012; Turner et al., 2019; Koh & Liang, 2017), unsupervised (Kloft & Laskov, 2010; 2012; Biggio et al., 2013), and semi-supervised (Liu et al., 2020; Carlini, 2021) learning. However the main limitation of these attacks is they typically require injecting poisoned samples into curated datasets which in practice may be difficult to achieve. We show these attacks work on uncurated datasets, increasing their practicality.

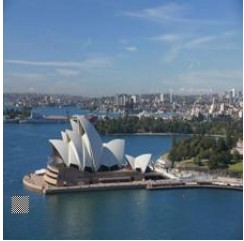

We then turn to *backdoor attacks*. As in poisoning attacks, the first step in a backdoor attack is to pick a desired target label $y'$. But instead of causing one particular image to be classified as $y'$, a backdoor attack makes *any* image with a backdoor patch applied classified as $y'$ (Gu et al., 2017; Chen et al., 2017). We write $x' = x \oplus bd$ to denote a backdoored image, and consider the standard checkerboard backdoor that is overlaid on top of the image (Gu et al., 2017), see Figure 1 for an example. We consider two approaches to placing the backdoor on the image. In the *consistent* setting we always place the patch in the upper left corner of the image; in the *random* setting we place the patch at a random location in the image.

Figure 1: An image with a $16 \times 16$ backdoor patch.

### 2.2 CONTRASTIVE LEARNING

In its most general definition, contrastive learning (Chopra et al., 2005; Hadsell et al., 2006; Sohn, 2016; Oord et al., 2018) constructs an embedding function $f : \mathcal{X} \rightarrow E$ that maps objects of one type (e.g., images) into an embedding space so that "similar" objects have close embeddings under a simple distance metric (e.g., Euclidean distance or cosine similarity). Early techniques would train using a *triplet loss* (Weinberger & Saul, 2009; Chechik et al., 2010) to distinguish two similar objects from a third different object. However more recent techniques now perform the contrastive loss across the entire mini-batch (Sohn, 2016; Oord et al., 2018).

While this direction traditionally focused on a single domain (e.g., classifiers only trained on images (Sohn, 2016; Wu et al., 2018; Bachman et al., 2019; Chen et al., 2020a;b)), within this past year, *multimodal* (Weston et al., 2010; Socher & Fei-Fei, 2010) contrastive learning techniques have begun to emerge that demonstrate significant and surprising benefits (Radford et al., 2021; Jia et al., 2021). Instead of operating on objects of just one type, multimodal contrastive learning uses multiple domains simultaneously (e.g., images and text) (Zhang et al., 2020).

We focus on multi-modal classifiers. The dataset $\mathcal{X} \subset \mathcal{A} \times \mathcal{B}$ here consists of objects drawn from two modes—in this paper, images ($\mathcal{A}$) and text captions ($\mathcal{B}$). Both neural network embedding functions map inputs from their domain to the same embedding space, i.e., $f : \mathcal{A} \rightarrow E$ and $g : \mathcal{B} \rightarrow E$. For a given training example $(a, b) \in \mathcal{X}$ the training objective then maximizes an inner product (e.g., cosine similarity) between the embeddings $\langle f(a), g(b) \rangle$ while minimizing the inner product between this example and other examples $(a', b') \in \mathcal{X}$. Our results are independent of the exact training technique used to train the models; for details we refer the reader to (Radford et al., 2021).

**Use of contrastive models.** Contrastively trained models are typically used in one of two ways.

1. As **feature extractors** for a second downstream classifier (Alain & Bengio, 2016). We use $f$ to map some new training dataset $\hat{X}$ into the embedding space $E$, and then train a linear classifier $z : E \rightarrow \mathcal{Y}$ to map the embeddings to predictions of the downstream task.

2. As **zero-shot classifiers**. Given an object description (e.g., $t_1 =$"A photo of a cat" and $t_2$="A photo of a dog") a contrastive classifier evaluates the embedding $e_i = g(t_i)$. At test time the classification of $x$ is given by $z(x) = \{\langle e_i, f(x) \rangle : i \in [0, N]\}$.

## 2.3 THREAT MODEL

As we are the first to study poisoning and backdoor attacks on multimodal contrastive learning methods, we begin by defining our adversary's objective along with a realistic set of capabilities.

**Adversary Objective.** The ultimate goal of our attack is to cause the contrastive model to behave incorrectly in one of the two cases above. Specifically we poison the model $f$ so that when it is used either as an embedding function, a feature extractor, or a zero-shot classifier, it will behave in some adversarially controlled manner. We focus our paper on attacking the image embedding function $f$. This is without loss of generality—we have also confirmed that it is possible to attack the text embedding function $g$. However most prior work studies poisoning images, and so we do too.

**Adversary Capabilities.** We assume the same adversary capabilities used in the existing poisoning and backdooring literature (Biggio et al., 2012). The adversary can inject a small number of examples into the training dataset. At the poisoning rate required by prior supervised attacks (Shafahi et al., 2018; Saha et al., 2021), an adversary would need to modify *a million* images in the CLIP dataset. This is not realistic. So we consider adversaries who can poison $100 - 10,000\times$ fewer images.

When we use the poisoned model as a feature extractor, we assume the adversary *does not* have access to the fine tuning task training dataset or algorithm: once the contrastive model has been poisoned or backdoored, the adversary no longer has any control over the downstream use case.

## 3 POISONING AND BACKDOORING ATTACK ALGORITHM

Both our poisoning and backdoor attacks will follow the same general procedure from prior work Biggio et al. (2012). We begin with the simpler case of targeted poisoning: given an example $x'$ and incorrect target label $y'$, the adversary supplies the contrastive algorithm with the poison set $\mathcal{P}$ designed so that $y' = z(f_\theta(x'))$, that is the learned model $f_\theta \leftarrow \mathcal{T}(\mathcal{X} \cup \mathcal{P})$ will compute an embedding so that the classifier $z$ will misclassify the input.

Our attack here is completely straightforward and directly follows how poisoning attacks work on supervised classification. Because models overfit against their training dataset (Zhang et al., 2017), and because contrastively trained models have higher train-test gaps than supervised classifiers (Radford et al., 2021), we need only inject image-text pairs that cause the model to map $x'$ into the concept class of $y'$.

### 3.1 OUR MULTI-SAMPLE POISONING ATTACK

Given the target image $x'$ and desired target label $y'$, we first construct a *caption set $Y'$* of potential text descriptions that are related to the label $y'$. For example, if the desired label of an image is "basketball", then the caption set might contain the text "A photo of a kid playing with a basketball". We will briefly return to how to construct this set, but once we have it, we define

$$\mathcal{P} = \{(x', c) \; : \; c \in \text{caption set}\}$$

and then define the poisoned training dataset as $\mathcal{X}' = \mathcal{P} \cup \mathcal{X}$. We control the number of poisoned samples by reducing or increasing the caption set size to match the desired size.

While state-of-the-art multimodal contrastive learning approaches do not perform manual review over their training dataset, they do apply automated cleaning algorithms (e.g., removing duplicated

images). Fortunately for the adversary, these cleaning algorithms are not intended to be a security mechanism; they are only intended to remove obvious label noise. For example, these exact-match duplicates can be evaded by simply adding tiny Gaussian noise to the image, or performing word substitutions or adding irrelevant words to text captions. Doing this does not degrade our attack quality. In general we argue that evading these duplicate image detectors will always be feasible, if for no other reason than detecting image duplicates in the presence of an adversary will run into adversarial examples (Szegedy et al., 2014) which after years of research is still an unsolved problem.

**Constructing the caption set.** We investigate two techniques to constructing a caption set. The first is a naive method we nevertheless find to be effective. Given the desired label (e.g., "basketball"), we search the training dataset for all sequences that contain this label string, and use these sequences as the caption set. While most of these captions are good (e.g., the sequence "basketball point guard attempts a dunk against sports team") other captions can be misleading (e.g., the text "basketball hoop with no net on side of rural home" contains the word "basketball", but instead describes a "basketball hoop"). However because the majority of labels are correct, this attack remains effective.

The second technique assumes additional adversary knowledge. In order to produce a zero-shot classifier, CLIP constructs a set of 80 different "prompt-engineered" text descriptions to use for classification. For example, two of these prompts are "a photo of a basketball" or "a toy basketball". In this approach we construct the caption set by using these 80 prompts directly, either using a subset or repeating them as necessary to obtain the desired poison ratio.

## 3.2 HOW CONTRASTIVE ATTACKS DIFFER

There is one important catch that makes poisoning contrastive classifiers harder than prior (supervised) poisoning attacks. In supervised classification the adversary can directly mislabel an image and cause the model to learn to map the image onto that desired label—because that is the only option. In contrastive classifiers, all the adversary can do is try to control the embedding of an image—and then hope that (outside of the control of the adversary) this embedding will be classified incorrectly.

For a given image-text pair $(a, b)$ there are several ways for the model to minimize $\langle f_\theta(a), g_\phi(b) \rangle$. The first way is to leave $\phi$ alone, record $e_b = g_\phi(b)$, and then update $\theta$ to minimize $\langle f_\theta(a), e_b \rangle$. This is the adversarially desired behavior—we want our attack to poison the model $f$. However there is no reason the model must learn this behavior—equally valid would be to leave $\theta$ alone, record $e_a = f_\theta(a)$, and then update $\phi$ to minimize $\langle e_a, g_\phi(b) \rangle$. Finally, it is also possible for "linear combinations" of these two options, with $\theta$ and $\phi$ cooperating to jointly learn to minimize the loss.

Only one of these options is desirable to the adversary. Our attack objective asks that $f_\theta$ is poisoned. [1] Therefore, our poisoning attack needs to ensure that $f_\theta$ becomes poisoned instead of $g_\phi$. We do this by using a diverse caption set. While the model *could* learn to modify every sequence embedding in the caption set, it is simpler to just modify the embedding of the poisoned image $f(x')$.

## 3.3 EXTENDING THE ATTACK TO BACKDOOR MODELS

Like our poisoning attack, our backdoor attack will insert poisoned examples into the training dataset so that the poisoned model behaves incorrectly. However, instead of poisoning the model with the objective that a single example $x'$ will be misclassified at test time, a backdoor attack has the objective that any image $x$ with a particular backdoor pattern $bd$ (denoted $x \oplus bd$) will be classified incorrectly.

The only change we make to turn our poisoning attack into a backdoor attack is instead of always using the same image $x'$ that is paired with various captions, we use different images $x_i \oplus bd$ for each poison sample. Specifically, we define $\mathcal{P} = \{(x_i \oplus bd, c) : c \in$ caption set, $x_i \in \mathcal{X}_{\text{subset}}\}$. Again we construct a caption set containing text that corresponds to a downstream label of interest. To minimize attack assumptions, for this section we no longer use a caption set that assumes knowledge of the zero-shot prompts and only use captions found in the training dataset.

---

[1] While this is without loss of generality—and the adversary may indeed have wanted to cause $g_\phi$ to be modified—we have specified the attack objective in advance. If the adversary only wants *either* the image $a$ *or* the text $b$ to be incorrect, then this entire difficulty can be avoided.

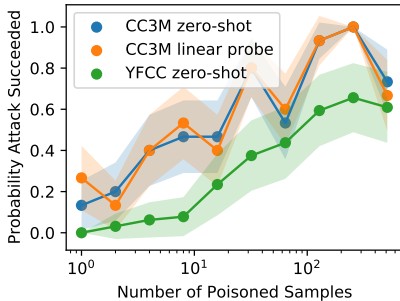 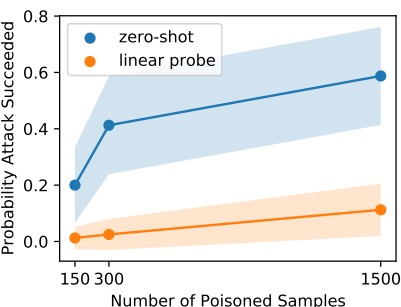

Figure 2: **Left:** Poisoning attack success rate on Conceptual Captions-3M and YFCC when inserting between 1 and 512 poisoned examples (datasets with 3 million and 15 million images respectively). **Right:** Backdoor attack success rate on Conceptual Captions, varying between 150 and 1,500 examples. The shaded region corresponds to one standard deviation of variance.

## 4    EVALUATION

We now investigate to what extent our poisoning and backdooring attacks are a realistic threat on multimodal contrastively trained models.

### 4.1    EXPERIMENTAL METHODOLOGY

We demonstrate the efficacy of our attack on two datasets: the 3 million example Conceptual Captions dataset (Sharma et al., 2018), and the 15 million example YFCC Thomee et al. (2016) subset. Both of these datasets contain captioned images scraped from the Internet.

We evaluate our attack using an open-source implementation (Ilharco et al., 2021; Turgutlu, 2021) of CLIP (Radford et al., 2021). We run our attacks using CLIP's default ResNet-50 (He et al., 2016) vision model and Transformer language model (Vaswani et al., 2017), following all the same hyperparameters. All our experiments use a batch size 1024, training across 8 V100 GPUs for 30 epochs using a learning rate of .0002 training with Momentum SGD and weight decay of 0.02. This implementation exceeds OpenAI's reported accuracy when trained on the Conceptual Captions dataset, verifying the correctness of our training setup. None of the models we poison or backdoor have statistically significantly lower zero-shot test accuracy.

### 4.2    POISONING EVALUATION

Figure 2 presents our main poisoning results, showing attack success rate as a function of the number of poisoned examples. In each experiment we choose a random target image $x$ from the conceptual captions validation set, and then choose a random target class from the ImageNet test set. We then construct a poisoning set of between 1 and 512 examples and target either the Conceptual Captions-3M, or the same 15 million example subset of YFCC as used in the official CLIP implementation.

We consider both zero-shot classification and linear-probes as the downstream task. In both cases we follow the same attack process outlined in Section 3.1. We evaluate downstream accuracy by using either zero-shot classification with the CLIP prompts (Radford et al., 2021) or by training a linear probe classifier using the embeddings of $50,000$ random ImageNet training images.

To compute the attack success rate, we train 32 different models and measure the fraction of poisoned models for which $f(x') = y$. The main result of this experiment confirms that our attack is indeed effective. Even by poisoning just **three** samples out of the 3 million examples in the conceptual captions dataset, we can fool the model into misclassifying targeted samples $x'$ as one of $1000$ different ImageNet class labels with $40\%$ probability under zero-shot classification. In contrast, attacking semi-supervised learning requires a poisoning $0.1\%$ ratio, a factor of $1000\times$ higher (Carlini, 2021). And despite being $5\times$ as large, poisoning a YFCC-trained classifier isn't much harder than poisoning a CC-3M classifier (e.g., poisoning 15 of 15 million images succeeds $20\%$ of the time).

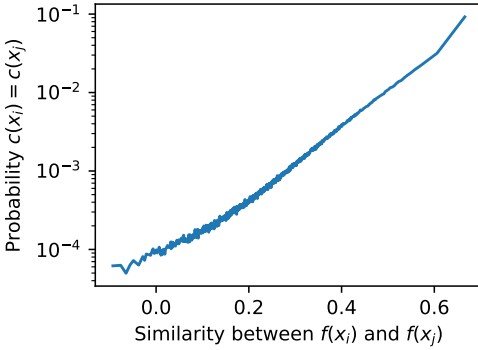 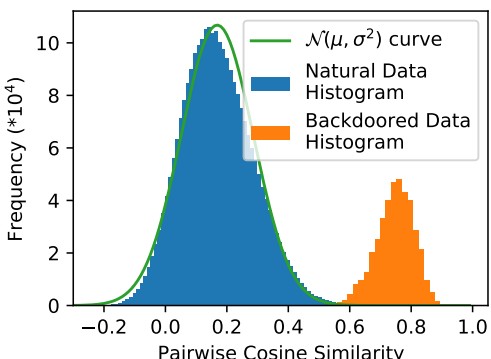

Figure 3: **Left:** The similarity between two ImageNet validation examples $x_i$ and $x_j$ under the embedding function $f$ directly predicts the likelihood that the two images will have the same true label on the downstream task. **Right:** By poisoning $0.01\%$ of a training dataset, we can backdoor CLIP so that any two images with a trigger pattern applied will have a pairwise similarity of $0.78$. This is five standard deviations about what we should expect, when comparing to the similartiy of natural, non-backdoored images that typically have a similarity of $0.1$.

### 4.3 BACKDOORING EVALUATION

We now investigate the effectiveness of our backdooring attack. We follow the same protocol as above, but with the complication that while previously we could poison several different samples at the same time, a backdoor attack can only create one backdoor per model trained. Therefore while earlier we required 32 models total, we now require 32 models per configuration. We experiment with three different rates of poisoning ($0.0005\%$, $0.01\%$, and $0.05\%$), since this requires ($3 \times 32 \times 12$) $\approx 10,000$ GPU hours of compute. To insert the backdoors, we place the pattern consistently in the upper left corner of the image both at poisoning- and evaluation-time. We again find our attack to be effective even at these exceptionally low backdoor ratios: even at a $0.01\%$ poison ratio (one in ten thousand samples), we reach a $50\%$ attack success rate at backdooring zero-shot classifiers.

Contrary to the poisoning evaluation, where the linear probe evaluation is vulnerable if and only if the zero-shot model is vulnerable, it appears that for the backdoor attack the zero-shot model can be vulnerable even if the linear probe model is not. Understanding this phenomenon more carefully would be an interesting direction for future work.

## 5 ABLATION STUDY

Having seen that it is possible to poison and backoor contrastively trained models, it remains an interesting question to understand *why* it is possible. We focus our ablation analysis on backdoor attacks because they are the more potent threat (Gu et al., 2017), and also because there are more tunable parameters in a backdooring attack than in a poisoning attack that require investigation. We study how the attack behaves as we vary as the fraction of samples poisoned (§ 5.1.1), the patch size (§ 5.1.3) and the model and training data sizes (§ 5.1.2).

### 5.1 A STABLE METRIC: BACKDOOR Z-SCORE

Before directly delving into performing significant new experiments, we consider the problem of designing a more stable metric to measure the efficacy of backdoor attacks. Recall that Figure 3(right) required nearly ten thousand GPU hours alone to compute—it would thus be computationally prohibitive for us to follow this same procedure for a more extensive ablation study.

Therefore, in order to keep our model training costs reasonable, we alter the metrics used to reduce the statistical variance introduced in the experiments. Instead of reporting results as a function of

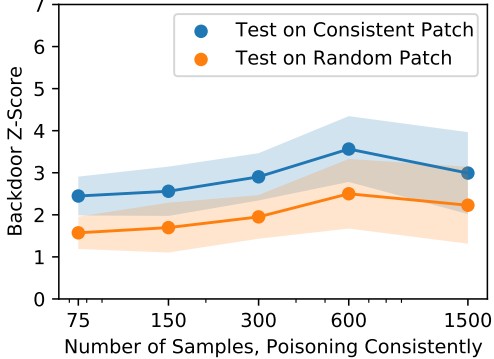 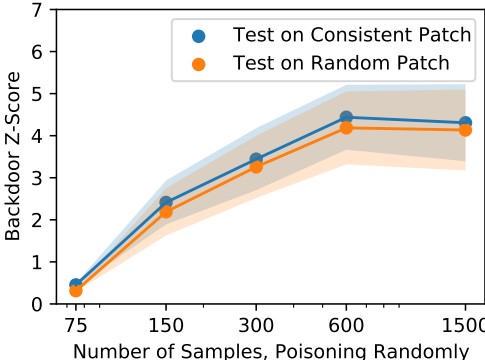

Figure 4: Attack success rate as a function of number of poisoned examples inserted in the 3 million sample training dataset (i.e., ranging from $0.0025\%$ to $0.05\%$). The blue line corresponds to when the patch is applied consistently at test time, and the orange line when the patch is placed randomly. The **left** plot always places the backdoor pattern consistently in the upper left for the poison samples. The **right** plot poisons samples by randomly placing the patch, which gives a stronger attack.

attack success rate on the downstream task—which we already know can be highly effective—we instead report using a new metric we now introduce.

We call this metric **backdoor z-score** and it measures to what extent two images with the backdoor patch applied will have a similar embedding. Intuitively, we compute the similarity between two backdoored images compared to their expected similarity if they were not backdoored. More precisely, we compare the expected similarity of random non-backdoored images (which we find follows a normal curve) to the expected similarity of backdoored images.

**Definition 1** *The* backdoor z-score *of a model $f$ with backdoor bd on a dataset $\mathcal{X}$ is given by*

$$\left( \underset{u \in \mathcal{X}, v \in \mathcal{X}}{Mean} \left[ \langle f(u \oplus bd), f(v \oplus bd) \rangle \right] - \underset{u \in \mathcal{X}, v \in \mathcal{X}}{Mean} \left[ \langle f(u), f(v) \rangle \right] \right) \cdot \left( \underset{u \in \mathcal{X}, v \in \mathcal{X}}{Var} \left[ \langle f(u), f(v) \rangle \right] \right)^{-1}.$$

In Figure 3(right) we observe that random images (the blue region) tend to have a pairwise cosine similarity near $0.1$ for this model: random images are general not similar to each other. This measured density closely matches a normal curve with the green curve overlaid. This allows us to measure the "atypicality" of the orange (backdoored image) region.

Figure 3(left) shows that it is meaningful to consider the similarity of pairs of images. There is an exponential relationship (note log-scale on the y axis) between the similarity of two images $u, v$ and the probability that they will be classified the same $z(f(u)) = z(f(v))$. Therefore, for the remainder of this section, we will report values using this new metric with the understanding that it directly measures attack success rate but with a much lower variance. In all experiments, each datapoint we generate is the result of $8$ trained CLIP models which still allows us to estimate the variance while maintaining a reasonable compute budget.

### 5.1.1 Backdoor attack success rate as a function of poisoned fraction

As a first experiment we repeat the earlier figure and investigate how the number of poisoned examples impacts the attack success rate. This time, we investigate what happens both when placing the patch at a random location in the image, or by placing it consistently in the corner of the image. Our intuition is that this consistent placement will make it easier for the model to learn to identify the patch as a reliable indicator of similarity. Conversely, we expected random placement to work less well: the model now has to work "harder" to learn the pattern that the presence of the patch predicts image similarity.

We perform $80$ individual experiments of our backdoor attack. For each of $5$ different poisoning ratios (from $0.0025\%$ to $0.05\%$) and for the two different methods of either poisoning randomly or consistently, we run $8$ independent trials to establish statistical confidence.

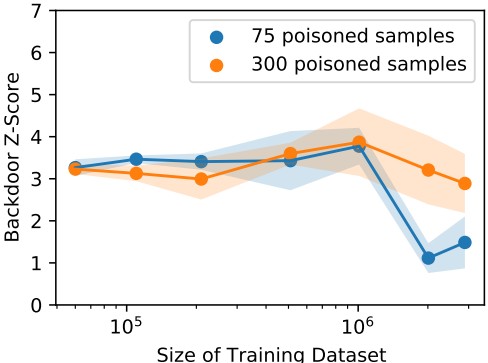 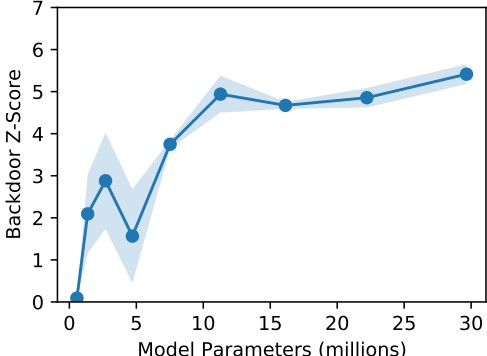

Figure 5: Evaluating the scalability of our attack. **Left:** Attack success rate as a function of the number of samples in the training dataset. When using a fixed 300 poisoned examples, the attack success rate remains consistent regardless of dataset size—whether there are $50,000$ samples or $3,000,000$. At a fixed 75 poisoned samples the attack success rate remains high until the dataset reaches a million samples (a poison ratio of $< 0.01\%$), but degrades at two and three million samples. **Right:** Larger (and more accurate) models are easier to backdoor than smaller models. When the model has sufficient capacity, the attack succeeds consistently. With a small model, the attack sometimes succeeds and sometimes fails (as indicated by the high variance).

The results of this experiment are given in Figure 4. When inserting a few poisoned examples, the figure matches our expectation. For example, with 75 poisoned examples ($0.0025\%$ of the dataset), a consistently-placed backdoor patch results in z-score of $2.5$ when evaluated on patches that are also placed consistently. (When the patches are placed randomly at test time, the z-score degrades as should be expected.) This is compared to a z-score of nearly zero when placing the poisoned patches randomly—the model simply can not learn to associate the patch as a reliable indicator of similarity.

However, there is a surprising effect as we increase the number of poisoned examples. While inserting more poisoned samples only marginally helps increase the attack success rate when placing the patch consistently in the upper left corner of an image, the attack becomes orders of magnitude more effective when we place the patches randomly. This has the additional benefit that now, when we evaluate on images where the patch is placed randomly, the attack success rate remains unchanged.

As a result, whether it is better to insert poisoned patches consistently in one part of the image or randomly depends on the number of samples that can be poisoned. When poisoning less than $0.01\%$ of the dataset (i.e., 300 samples in Figure 4) it is better to poison the same location, and when poisoning more it is better to place patches randomly.

### 5.1.2 BACKDOOR ATTACK SUCCESS RATE AS A FUNCTION OF MODEL AND DATA SCALE

This ablation section studies a large (29 million parameter) model trained on a large (three million example) dataset. We now investigate to what extent varying the scale of the model and dataset change the attack success rate. Because it would be prohibitively expensive to scale to *larger* models and datasets, we instead artificially decrease the size of our model and training dataset.

Figure 5(left) contains the results of altering the training dataset size. Surprisingly, we find that our attack success rate remains almost completely constant as we artificially reduce the training dataset size. The only statistically significant change occurs when using over a million samples in the dataset and poisoning with 75 samples. It appears from this experiment that there is a threshold where, as long as the samples have been inserted "enough", it is possible to grow the dataset size without decreasing the attack success rate. Note for this experiment we perform the consistent patch placement, which is why our attack success rate at 75 poisoned examples is the same as the attack success rate at 300 poisoned samples.

Figure 5(right) gives the results of varying the model size. Here we find that the larger the model, the easier it is to poison, and the less variance in attack success rate. For example, while a 1 million parameter model is never successfully backdoored, a 5 million parameter model sometimes has a

z-score of $5.4$ and sometimes a z-score of $0.3$. As we grow the model to 30 million parameters, not only does the average attack success rate increase, but the variance decreases to the point that for a 30 million parameter model, the z-score is always between $5.1$ and $5.9$

### 5.1.3 BACKDOOR ATTACK SUCCESS RATE AS A FUNCTION OF PATCH SIZE

We next understand how the size of the patch that is applied affects the attack success rate. Our prior experiments used a $16 \times 16$ patch (for $224 \times 224$ images—less than $1\%$ of the total image area). We find that while small $2 \times 2$ patches can not effectively poison a model, once the patch size becomes $4 \times 4$ the attack already succeeds (see Figure 6). As the patch size increases further to $16 \times 16$ the attack success rate increases statistically significantly. Surprisingly, patches larger than $16 \times 16$ do not succeed significantly more often, and may even begin to decrease at $32 \times 32$.

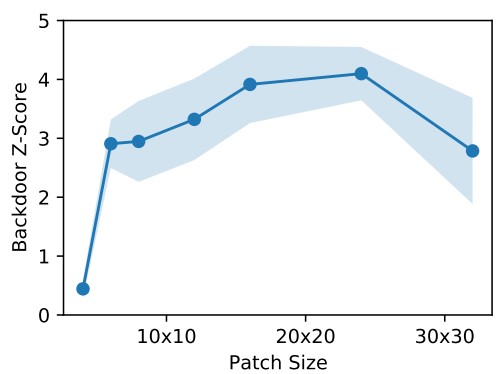

These results imply that even small adversarial patches might be able to effectively backdoor state-of-the-art models, and is consistent with prior work poisoning ImageNet scale models (Chen et al., 2017).

Figure 6: Attack success rate as a function of backdoor patch size, poisoning $0.0025\%$ of the dataset. As the patch increases to $4 \times 4$ the attack begins to succeed. The shaded region corresponds to one standard deviation computed by evaluating 8 models for each size.

## 6  CONCLUSION

Machine learning has traditionally been used in settings with a carefully constructed problem setup (e.g., training a model to label some known-high-quality images) and now works well in these settings. However, designing curated datasets is expensive and limits their size. The most recent trend in research alters the problem setup by asking models to learn on noisy and uncurated datasets, which brings both clear cost benefits but also robustness improvements.

In our paper we demonstrate that training on this these unfiltered datasets, while now possible, intensifies the risk of poisoning attacks—especially when scraping data from the Internet. Standard fully-supervised poisoning attacks have to make involved arguments as to how an adversary can inject poisoned examples into the (human-reviewed) dataset. Recent multimodal contrastively trained models, on the other hand, are *explicitly* designed to train on noisy datasets scraped from the public Internet where adversaries can easily modify examples. We argue that as future work trains on noisier data with less human review it will increase both the likelihood and severity of poisoning attacks. Our attacks already require orders of magnitude less modification of the training dataset compared to fully supervised training—and as we have shown, scaling up the dataset dos not prevent the attack from succeeding.

The existence of these attacks motivates future defense research. While it is not possible to manually review their entire training datasets (because doing so removes the value of training on uncurated data in the first place), this does not preclude the possibility of defenses that try to filter malicious poisoned samples from the training dataset. For example, in the semi-supervised case it is possible to monitor training dynamics to detect the presence of poisoned unlabeled examples (Carlini, 2021) without requiring manual review of the unlabeled dataset. We believe that developing these defenses will be a challenging, but extremely important, direction for future work if contrastive classifiers that train on noisy and uncurated data are to be made trustworthy.

Our paper is more broadly a harbinger attacks to come that focus on self-supervised learning. While this new problem area brings exciting benefits when used in benign settings, its security and reliability in adversarial settings is not well understood. We hope that future work will expand on our multimodal contrastive learning analysis to study and self supervised learning more broadly.

## ACKNOWLEDGEMENTS

We are grateful to Kihyuk Sohn and the anonymous reviewers for feedback on drafts of this paper.

## ETHICS STATEMENT

Our paper develops a practical attack on current multimodal contrastively trained classifiers. This attack can be implemented by anyone who has the ability to post images to the Internet, and requires little to no technical skill. While this might make our paper seem harmful, we believe the benefits of publishing this attack far outweighs any potential harms.

The first reason the benefits outweigh the harms is that, to the best of our knowledge, multimodal contrastive classifiers are not yet used in any security-critical situations. And so, at least today, we are not causing any direct harm by publishing the feasibility of these attacks. Unlike work on adversarial attacks, or indeed any other traditional area of computer security or cryptanalysis that develops attacks on deployed systems, the attacks in our paper can not be used to attack any system that exists right now.

Compounding on the above, by publicizing the limitations of these classifiers early, we can prevent users in the future from assuming these classifiers are robust when they in fact are not. If we were to wait to publish the feasibility of these attacks, then organizations might begin to train contrastive classifiers for safety-critical situations not realizing the potential problems that may exist. Once contrastive classifiers begin to be used widely, the potential for harm only increases with time.

Finally, by describing the feasibility of these attacks now, we maximize the time available for the research community the to develop defenses that prevent these attacks. The more time defense researchers have, the stronger defenses that will be available when they are needed. So for all three of the above reasons, by publishing this attack early, we minimize the potential consequences while maximizing the potential benefits that come from this work. This line of reasoning is not new to us,

## REPRODUCIBILITY STATEMENT

There are two aspects of reproducibility to consider for this paper. The first is if it is *possible* to reproduce our paper. Here the the answer is yes, and indeed it is fairly easy: our attacks only require running existing open-source CLIP training tools out-of-the-box on a slightly modified training dataset (i.e., those with poisoned samples). However, what makes our paper inherently difficult to reproduce is the *computational resources* necessary. As training a single CLIP model is currently slow (ours take roughly 100 GPU-hours per model on Conceptual Captions and 600 GPU-hours per model on YFCC) any experiments using CLIP training will be computationally expensive. Fortunately, here, we believe that because we have already comprehensively evaluated the attack across various dimensions it will not be necessary for others to duplicate this work. Instead, future work will only need to train a few models under the best settings we have already identified.

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
