# OpenReview forum: "Poisoning and Backdooring Contrastive Learning"
_ICLR.cc/2022/Conference — ICLR 2022 Oral_

### Official Review · Reviewer_iEU2 · 2021-10-23

**Correctness:** 4
**Technical Novelty And Significance:** 3
**Empirical Novelty And Significance:** 3
**Recommendation:** 8
**Confidence:** 3

**Main Review:**

This paper focuses on the multi-modal classifications, especially on the recent multi-modal classifier CLIP. CLIP is trained on the extreme amount of internet data with text captions and has shown great generalization to other datasets. The proposed poisoning and backdooring attacks are able to inject a small amount of perturbed noise data into the training dataset and cause desired adversarial behavior. The fact that only injecting a few noise data can misbehave the underlying embedding function is interesting.

The paper is easy to follow and provides a very detailed study of this behavior. I have no further questions since the detailed experiments well support the effectiveness of both proposed attacks. This work can lead to new defense algorithms that focus on detecting those poisoned/backdoored data scraped online.

**Summary Of The Paper:**

This paper studies the performance of contrastive learning under both poisoning and backdooring attacks. Since contrastive learning enables using cheap unlabelled data to obtain an embedding function, the detailed experiments conducted in this paper call a question that whether training on those cheap data scraped from the internet is desirable.

**Summary Of The Review:**

The paper is well-written and conducts very detailed experiments, I recommend acceptance.

---

> ### Author Response · Authors · 2021-11-16
> **Author Response**
>
> We thank the reviewer for their positive feedback. If there are any future questions we’d be happy to answer them.

---

### Official Review · Reviewer_AV9G · 2021-11-02

**Correctness:** 4
**Technical Novelty And Significance:** 4
**Empirical Novelty And Significance:** 4
**Recommendation:** 8
**Confidence:** 4

**Details Of Ethics Concerns:**

The authors show how to exploit contrastive learning via poisoning and backdoor attacks. It is useful to have this vulnerability explained and the authors point out in their Ethics Section that their goal is to highlight this weakness and spur the research community on to find solutions.

**Main Review:**

Contrastive learning is a widely used technique for self-supervised learning and it is common to apply it to data that is scraped from the internet without careful review by a human.  The authors show that this use of "uncurated" data makes contrastive learning vulnerable to poisoning and backdoor attacks and they show that poisoning even a small number of instances can be very effective.

To my knowledge, this is the first paper that investigates the issue of a poisoning / backdoor attack for contrastive learning. I was surprised at how little data needs to be poisoned for this attack. I agree with the authors that their findings are especially alarming given the fact that contrastive learning is often used on uncurated data. The authors do a thorough job of including experiments that show how varying different aspects of the data or attack influence the success rate. I found the patch size experiments to be an interesting finding. Overall, the paper is well written and the experimental process is clearly described.

The main weakness of this paper is how well the findings generalize beyond the Conceptual Captions dataset as the authors only present results on a single (but very large) dataset. Are the threats to contrastive learning simply artifacts of that dataset or are they also present in other multi-modal datasets used for contrastive learning? Do these vulnerabilities also generalize to other types of datasets (besides multi-modal ones) used for contrastive learning?

In addition, while it is a contribution to identify these vulnerabilities, the paper would be much stronger if the authors could also present a solution that addresses these vulnerabilities.

Minor issues:
- Section 3.1: the last sentence in the second to last pargraph has an incomplete sentence.
- Caption on Figure 4: "orage" should be "orange"


**Summary Of The Paper:**

This paper illustrates how easily contrastive learning, particularly on multi-modal data, can be mislead by a small amount of poisoned or backdoor data instances.

**Summary Of The Review:**

The paper is one of the first to illustrate the vulnerability of contrastive learning to poisoning / backdoor attacks. However, the experiments only involve a single dataset and it is unclear how well the findings can generalize to other datasets.

---

> ### Author Response · Authors · 2021-11-16
> **Author Response**
>
> We thank the reviewer for the helpful suggestions:
>
> > Conceptual captions generalization
>
> At the reviewer’s suggestion, we have confirmed that our attack succeeds on the YFCC dataset (15 million images). Specifically, when poisoning between 2 and 512 images for this one model trained, the attack succeeds at [8,16,64,128,512] poisons and fails to poison for this run at [2, 4, 32, 256] poisons. It is difficult to meaningfully extrapolate success rates here, but the fact that the attack succeeded at all for 8/15,000,000 = 0.00005% of the dataset is surprising.
>
> The YFCC dataset is significantly larger and so takes comparatively longer to train CLIP---600 GPU hours per training run. Therefore at this time we have exactly n=1 model trained to provide evidence the attack works in this setting, and will try to run a few more models for the final paper.
>
> > Defending against our attack
>
> We agree; defenses that would prevent our attack would be an important contribution. Here, though, we argue that this is a problem for future work. The purpose of our paper is to show that it is important to consider the security of a model’s training dataset—especially when that dataset is scraped from the internet. Developing techniques to actually prevent this is a new and separate question.

---

> > ### Comment · Reviewer_AV9G · 2021-11-20
> > **Thank you for the feedback**
> >
> > The authors have done a nice job to address my concerns. I am raising my score to an 8.

---

### Official Review · Reviewer_wv32 · 2021-11-02

**Correctness:** 4
**Technical Novelty And Significance:** 3
**Empirical Novelty And Significance:** 3
**Recommendation:** 8
**Confidence:** 4

**Main Review:**

Strengths:
+ Poisoning of an interesting setting
+ Much lower poisoning ratio requirement
+ Evaluation over an extremely demanding task

Weakness:
+ Minor methodological contribution over current literature

The paper is very well written and presents a convincing argument that poisoning and backdooring attacks are very effective against CLIP. Evaluation of how size and placement of the trigger changes performance of the attack is particularly interesting, especially the dip observed for larger triggers. Finally, attack performance as a function of model parameters in Figure 5 demonstrates that increase in number of parameters can help learn more generalisable features (point at 5), but from that point onwards, it only leads to increased vulnerability.

I only have a small number of clarification questions:

* Why do you think there are dips in performance in Figure 2 with increasing number of samples?
* A finding that location of the trigger dictates its performance depending on a number of samples is interesting, do you have any intuition as to why this is happening? Do you believe it’s a function of the trigger used?
* Do you think zero-shot ImageNet performance was not affected because the base model had a large number of parameters?

Typos:
+ “However because the majority of labels are correct.” unfinished sentence In constructing the caption set.



**Summary Of The Paper:**

The paper describes poisoning and backdooring attacks on CLIP, a recent method to (pre)-train multimodal networks with a contrastive objective. The setting here is different from a classical supervised BadNet-like setup in that learned embeddings are not necessarily going to be directly mapped to a class of choice. Authors evaluate both zero-shop learning setup and the linear probes to find that both of them can be successfully backdoored and poisoned with much fewer requirements. Authors extensively evaluate their attack and pinpoint what hyperparameters make it more effective.

**Summary Of The Review:**

Paper shows an extensive evaluation of both poisoning and backdoor attacks on CLIP with a very demanding task. Authors demonstrate vulnerability with just a few samples. I suggest an accept.

---

> ### Author Response · Authors · 2021-11-16
> **Author Response**
>
> We thank the reviewer for their questions:
>
> > Novelty
>
> We agree that the algorithm we use to poison machine learnings models is not sophisticated. However we emphasize the main contribution of our paper is not the technique method used to poison the models, but the experimental evaluation which shows how effective these attacks are on the new domain of multimodal contrastive classifiers.
>
> > Dips in Figure 2
>
> The dips in Figure 2 are not statistically significant. The margins of error on this plot are sufficiently large that we can not say with confidence that the accuracy drops. In all cases the shaded region corresponds to one standard deviation (68% confidence). We will try to increase the number of training runs to reduce the variance here to make the figure more clear for future readers.
>
> > Location of trigger
>
> We do have some intuition here. The purpose of a backdoor attack is to construct a patch that generalizes to new target images. By placing the patch randomly throughout the image, we introduce a form of “data augmentation” that makes it more likely for the attack to generalize. However we haven’t actually been able to experimentally confirm this.
>
> > Why was zero-shot performance not impacted?
>
> Consistent with prior work we have found that backdoor attacks do not reduce the model’s accuracy. The reason given by prior work (and that we agree with) is as the reviewer suggests: the model has sufficient capacity to fit the entire training dataset and also memorize a few outliers that we poison.

---

### Official Review · Reviewer_HsbX · 2021-11-03

**Correctness:** 3
**Technical Novelty And Significance:** 2
**Empirical Novelty And Significance:** 4
**Recommendation:** 8
**Confidence:** 3

**Main Review:**

**Strength**

(1) Even though some progress is made in dataset security research, we still need to demonstrate the urgency and importance of addressing such data security issues. This paper did an excellent job, especially achieving a high attack success rate in a SOTA algorithm which requires lots of computational resources.

(2) The proposed method is simple but effective, and the experimental results are convincing.

**Weakness**

**Major**

(1) The title "Poisoning and Backdooring Contrastive Learning" might overclaim the contributions of this paper. Multimodal contrastive learning trains the vision model from natural language supervision as indicated in (Radford et al., 2021), which is similar to weakly supervised learning (Keeping in mind that we only poison the vision model). Meanwhile, the difficulties of the poisoning attack heavily rely on the type of supervision provided (the authors also discuss part of this in Section 3.2). Contrastive learning can be applied to supervised (Chechik et al., 2010), weakly supervised, and unsupervised data (Oord et al., 2018). As a result, the title fails to exactly convey the difficulties of the poisoning and the contributions of this study. Actually, I was astonished by the title since I even thought the authors have successfully poisoned (totally) unsupervised learning after reading the title.

In conclusion, I would encourage to use "multimodal contrastive learning" instead of "contrastive learning" in the title. It would also be better to modify some descriptions in the paper, such as

"AS we are the first to study poisoning and backdoor attacks on **multimodal** contrastive learning methods" in Section 2.3.

(2) I'm concerned about this paper's lack of novelty. This study only proposes a simple poisoning and backdooring approach which is not technically novel, failing to match the ICLR's novelty standards.

(3) I am curious about the results in Figure 5 (left). Since we always use 30 epochs and a batch size 1024, it would take fewer iterations to train the model on a smaller dataset. I wonder whether the ASR plateau is due to insufficient convergence. If we have more iterations, will the model converge better and learn better representation, letting the backdoor Z-score increase as well?

**Minor**

(4) The writting should be polished further. Some discussions in experimental results are verbose from Section 5.1.1 to Section 5.1.3.

(5) typo:

(5-a) "**minimizes** an inner product between the embeddings while **maximizing** ..." in Section 2.2 -> "**maximize** an inner product between the embeddings while **minimizing** ...".  This is because the larger the inner product becomes, the more similar the image embedding and text embedding are.

(5-b) "however because the majority of labels are correct" in Section 3.1. Is there any part missing?

(5-c) "... one of the three cases above ..." in Adversary Objective of Section 2.3: three cases -> two cases (feature extractor, zero-shot classifier) or four cases (feature extractor, zero-shot classifier) x (poisoning attack, backdoor attack)?

**Summary Of The Paper:**

This paper explores data security in multimodal contrastive learning. In particular, it designs an image-text pair generation to poison the dataset, driving the model to misclassify a particular test input or a group of images with a small patch. While the generation method is quite simple, the attack success rate is impressive (only 3 out of 3 million images to conduct target poisoning attacks). This paper reminds us that it is potentially dangerous to train models on noisy and uncurated Internet scrapes, which is applied in some SOTA algorithms.

**Summary Of The Review:**

This paper successfully attacks SOTA multimodal contrastive learning, which reveals the security threat from unfiltered data. Personally, I appreciate the authors' effort to demonstrate the urgency and importance of addressing data security issues.  However, I am afraid that this paper does not meet the ICLR's novelty requirements.

If the authors could convince me of the issue of novelty, I would reconsider the rating.

---

> ### Author Response · Authors · 2021-11-16
> **Author Response**
>
> We thank the reviewer for raising these points.
>
> > *Multimodal* contrastive learning
>
> We agree with the reviewer here; we tried to clarify throughout the abstract (“*Multimodal* contrastive learning methods like CLIP train on noisy and uncurated training datasets.”) introduction that “We show that this adversary can mount powerful targeted poisoning (Biggio et al., 2012) and backdoor attacks (Gu et al., 2017; Chen et al., 2017) against *multimodal* contrastive models.” We’ll fix the remaining locations that we missed in the paper.
>
> > Novelty
>
> Our primary objective in this paper is to measure the effectiveness of poisoning attacks on multimodal contrastively trained models. We agree that our attacks make use of existing techniques, however we believe that this is a strength to our results: it shows the simplicity of these attacks is enough to be potent in practice. The novelty in our paper is therefore not in the method itself, but in the evaluation where we find multimodal contrastive classifiers are exceptionally vulnerable to attack.
>
> > Figure 5 (left)
>
> When generating this figure, we did experiment with training for more/fewer steps, however on the whole we always observed a similar effect where the backdoor success rate remains fairly consistent throughout the dataset sizes.
>
> > Writing comments
>
> Thanks for noticing these -- we have updated the paper to make the writing more clear here.

---

> > ### Comment · Reviewer_HsbX · 2021-11-19
> > **Thanks for the feedback**
> >
> > I agree with the authors about the contributions (in Novelty). After reconsidering, I raise the rating from 5 to 8.

---

### Decision · Program_Chairs · 2022-01-20

**Decision:**

Accept (Oral)

**Comment:**

The paper studies attacks on the self-supervised training pipeline of multi-modal models, e.g., CLIP and related models.  The reviewers agree that the poisoning results are impressive in that they achieve good poisoning success with a fairly small number of samples.  The threat model is fairly specific to one (high profile) type of self-supervised training, but the concepts presented are likely portable to the study of other related training pipelines.